# Association between microbial water quality, sanitation and hygiene practices and childhood diarrhea in Kersa and Omo Nada districts of Jimma Zone, Ethiopia

Negasa Eshete Soboksa[1]*, Sirak Robele Gari[1], Abebe Beyene Hailu[2], Bezatu Mengistie Alemu[3]

1 Ethiopian Institute of Water Resources, Addis Ababa University, Addis Ababa, Ethiopia, 2 Department of Environmental Health Sciences, Jimma University, Jimma, Ethiopia, 3 College of Health and Medical Sciences, Haramaya University, Harar, Ethiopia

* yeroosaa@gmail.com

## Abstract

### Introduction

Diarrhea is one of the leading causes of child morbidity and mortality in low- and middle-income countries like Ethiopia. The use of safe drinking water and improved sanitation are important practices to prevent diarrhea. However, limited research has been done to link water supply, sanitation and hygiene practices and childhood diarrhea. Therefore, this study aimed at assessing the association between microbial quality of drinking water, sanitation and hygiene practices and childhood diarrhea.

### Methods

Community-based matched case-control study design was applied on 198 paired children from June to July 2019 in Kersa and Omo Nada districts of Jimma Zone, Ethiopia. Cases are children < 5 years of age with diarrhea during the two weeks before the survey. The controls are children without diarrhea during the two weeks before the survey. Twenty-five percent matched pair samples of water were taken from households of cases and controls. Data were collected using structured questionnaire by interviewing mothers/caregivers. A sample of water was collected in nonreactive borosilicate glass bottles and analyzed by the membrane filtration method to count fecal indicator bacteria. A conditional logistic regression model was used; variables with p-value less than 0.05 were considered as significantly associated with childhood diarrhea.

### Results

A total of 396 (each case matched with control) under-five children with their mothers/caregivers were included in this study. In the analysis, variables like presence of under-five child in their home (AOR = 2.76; 95% CI: 1.33–5.71), wealth status (AOR = 5.39; 95% CI: 1.99–14.55), main sources of drinking water (AOR = 4.01; 95% CI: 1.40–11.44), hand washing

**Funding:** This work was supported by Addis Ababa University, Ethiopian Institute of Water Resources. The funder had no role in study design, data collection and analysis, decision to publish, or preparation of the manuscript.

**Competing interests:** The authors have declared that no competing interests exist.

practice before water collection (AOR = 4.28; 95% CI: 1.46–12.56), treating water at household level (AOR = 1.22; 95% CI: 0.48–3.09), latrine use all the times of the day and night (AOR = 0.22; 95% CI: 0.06–0.78), using pit as method of waste disposal (AOR = 4.91; 95% CI: 1.39–13.29) and use of soap for hand washing (AOR = 2.89; 95% CI: 1.35–6.15) were significantly associated with childhood diarrhea. Moreover, 30% of sampled water from cases and 26% of sampled water from controls families were free from *Escherichia coli* whereas all sampled water analyzed for Total coliforms were positive.

## Conclusions

We conclude that the main sources of drinking water, hand washing before water drawing from a storage container, domestic waste disposal place and use of soap for hand washing were the most important factors for the prevention of childhood diarrhea.

## Introduction

Diarrheal disease is one of the leading causes of child morbidity and mortality in low- and middle-income countries like Ethiopia. It is most commonly caused by gastrointestinal infections, which kill around 2.2 million people globally each year, typically under-five children in developing countries [1]. A retrospective analysis of data from 145 countries shows that about 58% of diarrheal deaths are caused by unsafe water and poor sanitation [2]. Inadequate drinking water and sanitation are associated with considerable risks for diarrheal disease [3]. Poor practice of handling and treatment of drinking water, poor hand washing practices and inconsistent use of the toilet were associated with the occurrence of childhood diarrhea [4,5]. A study done in Ethiopia reported that safe child stool disposal is associated with decreased child diarrhea incidences by 23% [6].

Safe water supplied together with good sanitation and hygiene practices is more effective for reducing diarrhea even though the reduction of diarrhea [3,7]. Report of a recent study showed that improving household drinking water quality and changing people's behavior towards safe sanitation practices is an important mechanism to protect the risk of childhood diarrhea [6]. Improving microbial quality of water at point-of-use could reduce 39%, diarrhea risk; improving sanitation and hand washing promotion reduce the incidence of diarrhea by about 30% [7,8]. The effect of point-of-use chlorine treatment significantly improves the quality of stored water in intervention households and also reduces the risk of diarrhea by 25%-58% as observed from studies [9–12].

The microbial quality of drinking water is inherently linked to poor sanitation practices. In areas where poor standards of hygiene and sanitation are practiced, fecal pathogens are the most common source of drinking water contamination [13,14]. Poor hygiene and sanitation practices are responsible for the fecal contamination of water obtained from shallow wells [15]. Contamination of drinking water supplies can occur in the source water as well as in the distribution system after water treatment has already occurred [16]. These imply drinking water quality interventions at the household level have been shown to be effective in reducing diarrheal diseases [17].

In Ethiopia, 60–80% of the communicable diseases are resulting from unsafe and inadequate water supply and poor hygienic and sanitation practices [18]. To solve the problem related to water supply, sanitation and hygiene, the Ethiopian government work with

stockholders[19]. Although significant efforts have been made to improve the coverage of water supply and improved sanitation facilities, a large proportion of households are still practicing open defecation, lack safe water supply, and therefore, communicable diseases like diarrhea are the leading causes of under-five mortality in Ethiopia [18,20].

There have been significant studies examining the association of water supply, sanitation and hygiene practices on reducing childhood diarrhea[21–24]. However, limited research has been done to link water supply, sanitation and hygiene practices and childhood diarrhea through matched case-control studies to measure the association of these practices on improving child health outcomes. Therefore, this study was designed to assess the association of improved microbial water quality, sanitation and hygiene practices on childhood diarrhea reduction in Kersa and Omo Nada districts of Jimma Zone, Ethiopia. The results of this study could help policymakers, program planners, donors and concerned bodies to take appropriate measures. The study may also identify gaps for researchers and organizations working on community water supply and child health.

## Materials and methods

### Study area

The study was conducted in Kersa and Omo Nada districts of Jimma Zone, Oromia Regional State, Ethiopia. The zonal capital, Jimma Town, is located 357 km away from Addis Ababa in the southwest Ethiopia. The Zone extends between 7013'– 8056' North latitudes and 35049– 38038' East longitudes. The altitude of these districts ranges from 1740 to 2660 above sea level. Agriculture is the major source of economy and it includes mainly the growing of coffee and cattle rearing. According to 2011 Ethiopian Fiscal Year annual report of Jimma Zonal Health Office, the population of Kersa and Omo Nada Districts were 227,959 and 208,517, respectively. Of this population, about 81.65% residents of the Kersa districts and 71.7% residents of the Omo Nada district rely on improved drinking water sources in 2018. In this year, the improved latrine coverage of the districts was 40% for Kersa and 39% for Omo Nada [25].

### Study design and population

Community-based matched case-control study design was conducted from June to July 2019 in Kersa and Omo Nada districts of Jimma Zone, Ethiopia. The source population was all under-five children living in the two districts. Cases are children < 5 years of age with diarrhea in the preceding two weeks during a house to house survey. The controls are children without diarrhea during the two weeks before the survey. Cases and controls were matched by date of birth within 12 months, sex and kebeles of current residence.

### Inclusion and exclusion criteria

All under-five aged children that have diarrhea and their mothers reside as district member were included in cases. Whereas all under-five aged children that have no diarrhea and their mothers reside as district members were included in the controls. Diarrhea is defined as having three or more loose or watery stools in a 24 hour period, as reported by the mother/caretaker of the child. Children whose mothers are seriously ill and could not communicate to give information were excluded from the study.

### Sample size and sampling procedures

The sample size was calculated using Stata Version 14.0 statistical software (StataCorp LP, College Station, TX 77845, USA), assuming the probability of exposure among control group 0.34

(from previous study), odds ratio = 1.83 which is the Odds ratio of diarrhea among those not using protected drinking water sources [26], probability of type 2 error and the level of significance taken as 0.20 and 0.05 respectively, a ratio of case to control of 1:1; and the sample size generated to be 180 cases. Allowing a 10% non-response rate gives a total of 396 (198 pairs) for both cases and controls. One hundred ninety-eight pairs of children, matched on age and residence kebeles, were recruited as cases and controls. Fifty percent matched pair samples of water was taken from households in cases and controls.

First eleven kebeles (the smallest administrative unit in Ethiopia that consist of at least 500 households) were selected by simple random sampling. Then all the households with under-five children were identified by house-to-house survey and cases and controls were assigned proportionally to each kebeles and were randomly selected. For every case selected randomly from giving kebeles, the controls are selected randomly from the same kebeles. Then 198 pairs of children, matched on sex, age and kebeles, were recruited as cases and controls. Respondents were mothers or caregivers of under-five children. If there were more than one under-five child in a household, the youngest one was included in the study.

## Data collection methods

Data were collected using pre-tested structured questionnaire by interviewing mothers/caregivers and water sample was collected from 25% of interviewed mothers/caregivers. For face-to-face interview questionnaire was developed and was administered through face-to-face interviews in the local language (Afan Oromo). These collected data included socio-demographics, sources of water for drinking and domestic uses, water storage practices, and water treatment techniques used in their home. The collected data also included hygienic and sanitation practices by asking questions related to hand washing practices at critical times (after defecation, before handling food/water, before feeding a child, after cleaning child stool), covering drinking water storage, cleaning utensils regularly before filling drinking water, touching / dipping fingers in water during collection, place of defecation, presence of functional latrine, type of latrine, disposal system of children feces and method of domestic waste disposal. Respondents were also asked about their encounters and experiences with diarrhea in their households.

The samples of water were collected from storage containers used for drinking purposes (at point of use). The method of water sample collection was according to the WHO guidelines for drinking water quality assessment [27]. Samples of water were collected in nonreactive borosilicate glasses that have been cleansed and rinsed carefully by wearing gloves to minimize potential contamination, given a final rinse with distilled water, and sterilized. For bacteriological analysis, the samples were immediately transported in ice-packed cooler boxes to the laboratory of Jimma University Environmental Health department. In the laboratory, it was analyzed for indicator bacteria. To determine the degree of contamination, Total Coliforms and *Escherichia coli* groups were determined using the membrane filtration technique as outlined by the APHA/AWWA/WEF [28]. The water samples were filtered through the sterile membrane filter and then placed face upward on an absorbent pad saturated with M-Lauryl Sulfate Broth. After incubation for 4 hours at 30˚C followed by 18 hours at 36˚C for Total Coliform organisms and 4 hours at 30˚C followed by 14 hours at 44˚C for Escherichia coli, the yellow colonies formed were counted for both and the results were calculated and expressed in colony forming unit (CFU) per 100 ml sample.

## Data processing and analysis

The data were checked, coded and entered using SPSS version 24.0 (IBM Corp., Armonk, NY, USA). Descriptive analysis was computed using SPSS and conditional logistic regression

analysis was done by Stata version 14.0 (StataCorp, College Station, TX, USA). Wealth index quintiles (poor, middle and rich) were generated with a statistical procedure known as principal component analysis (PCA). A conditional bivariate and multivariable logistic regression were used in the analysis. A conditional bivariate logistic regression was used to identify the association of outcome variable with each explanatory variables. Conditional multivariable logistic regression was used to adjust for confounders. All explanatory variables associated with the outcome variable in the conditional bivariate analysis with a p-value of 0.25 were selected and analyzed for adjustment. The odds ratio and 95% confidence interval (CI) was used to determine the effect of potential associated variables on the childhood diarrhea, which was considered as the outcome variable and to control confounding factors. All variables included in the final model and resulted with a p-value < 0.05 were considered as statistically significant association with childhood diarrhea.

### Ethical considerations

This study was conducted after Institutional Review Committee of Addis Ababa University, Ethiopian Institute of Water Recourses was approved the use of oral consent because of the high proportion of illiteracy to read the consent form. This oral consent was recorded if the study participants agreed to participate in the study after the purpose of the study was explained by data collector before data collection. When children with diarrhea were found during data collection, the data collector advise their mothers/caregivers about homemade therapy or they bring the child to the health center.

## Results

### Socio-economic and child factors

A total of 396 (each case matched with controls) under-five children with their mothers/caregivers was included in this study. The mean age of mothers/caregivers was 29.6(SD±5.6) for cases and 28.5(±4.5) for controls. Almost all the participants were Oromo in their ethnicity and Muslim in their religion both in the cases and controls. Regarding the marital status of the interviewed mothers/caregivers, about 97% cases and 91% controls were married. About 87% of cases and 81% of controls of this study participants were housewives. The majority of the interviewed mothers/caregivers: 58.6% of cases and 76.8% of the controls, had only one under-five child. Among the mothers/caregivers included in this study, 49% mothers/caregivers of cases and 44.9% of mothers/caregivers of controls had no formal education. About 45.5% of interviewed mothers/caregivers of cases and 21.2% of interviewed mothers/caregivers of controls were ranked as economically poor, whereas 25.8% of cases and 40.4% of control mothers/caregivers were ranked as wealthy.

Regarding indexed children, about 55% cases and 55% controls were male. The mean age of the children was 26.41 (±11.58) months for cases and 25.66 (±10.36) months for controls. About 52% cases and 68.2% controls of this study were partially breast-fed during the study period. The majority of these children, 85.4% cases and 83.3% controls started complementary feeding at the age of six years. In addition to that, the majority of the children has received measles and Rota vaccines (Table 1).

Table 1 shows the bivariate analysis of diarrhea status and socio-economic and child characteristics. In the analysis, the presences of under-five children in the house, current breast-feeding status, age at which child started complementary feeding and wealth status of the mother/caregiver were significantly associated with diarrhea. The remaining variables were not significantly associated with the outcome variable. The odds of having diarrhea among under-five children were higher in households whose family size was greater than five, but the

**Table 1. Socio-economic status and child characteristics associated with childhood diarrhea in Kersa and Omo Nada districts of Jimma Zone, Ethiopia.**

| Variables | | Cases | Controls | Crude OR (95% CI) |
|---|---|---|---|---|
| | | No (%) | No (%) | |
| Family size | Less than 5 | 106(53.5) | 96(48.5) | 1 |
| | 5 and above | 92(46.5) | 102(51.5) | 1.22(0.82–1.83) |
| Number of under-five in the household | 1 | 116(58.6) | 152(76.8) | 2.5(1.56–4.01)* |
| | 2 and above | 82(41.4) | 46(23.2) | 1 |
| Mothers age | ≤29 | 104(52.5) | 99(50.0) | 1 |
| | >29 | 94(47.5) | 99(50.0) | 1.10(0.75–1.62) |
| Highest level of education completed by the mother/ caregiver | Have no formal education | 97(49.0) | 89(44.9) | 0.76(0.42–1.38) |
| | Primary | 65(32.8) | 68(34.3) | 0.89(0.49–1.61) |
| | Secondary and above | 36(18.2) | 41 (20.8) | 1 |
| Occupation of mother/caregiver of the child | House wife | 172(86.9) | 160(80.9) | 1 |
| | Merchant | 22(11.1) | 29(14.6) | 1.44(0.79–2.67) |
| | Government employee | 4(2.0) | 9(4.5) | 2.39(0.73–7.81) |
| Current breast feeding status | Exclusive breast feeding | 3(1.5) | 2(1.0) | 1.02(0.16–6.38) |
| | Partial breast feeding | 103(52.0) | 135(68.2) | 2.07(1.33–3.21)* |
| | Not breast feeding | 92(46.5) | 61(30.8) | 1 |
| Age at child start complementary feeding (Months) | Three | 3(1.5) | 3(1.5) | 1.53(0.29–8.23) |
| | Four | 10(5.1) | 21(10.6) | 2.33(1.03–5.24)* |
| | Six | 169(85.4) | 165(83.3) | 1 |
| | I didn't remember | 16(6.6) | 9(3.5) | 0.53(0.23–1.26) |
| Received measles vaccination | Yes | 171(86.4) | 175(88.4) | 1.13(0.57–2.27) |
| | No | 22(11.1) | 21(10.6) | 1 |
| Received Rota virus vaccination (Rvv1, Rvv2) | Yes | 169(85.4) | 175(88.4) | 1.31(0.72–2.39) |
| | No | 29(14.6) | 23(11.6) | 1 |
| Wealth status | Poor | 90(45.5) | 42(21.2) | 1 |
| | Middle | 57(28.8) | 76(38.4) | 3.28(1.83–5.87)* |
| | Rich | 51(25.8) | 80(40.4) | 3.59(2.01–6.08)* |

*Statistically significant at P < 0.05

increment of the risk was not statistically significant. Risk of exposure to diarrhea among under-five children living with mothers/caregivers who attended primary school was reduced by 11% compared to those that had secondary education and above (crude OR = 0.89; 95% CI: 0.49–1.61). Similarly, the odds of having diarrhea was significantly higher among children living with mothers/caregivers grouped as wealthy compared to those grouped as relatively poor (crude OR = 3.59; 95% CI: 2.01–6.08) (Table 1).

## Water supply related factors

Concerning main sources of drinking water, families of 70.2% of cases and 78.3% of controls collected water from protected well/spring. The majority of the mothers/caregivers: 90.1% cases and 91.9% controls, informed that the water was not available from these sources when needed. The families of about 7.1% of cases and 4.5% of controls depended on unprotected sources like well, spring and river for cooking and washing. Almost all the study respondents were living in the residential environment less than one kilometer away from drinking water sources. The mean time taken to go there, get water, and come back from these sources of water was 32.96 (±21.18) minutes for cases and 31.42 (±20.46) minutes for controls families.

In this study, the average daily water consumption for cases were 12.41 (±5.72) liters and for controls were 13.82 (±6.30) liters. Most of the study respondents reported that this amount of water was no sufficient for their household. Mothers/caregivers of about 60% of cases and 69% of controls said that the main reason for inability to access sufficient quantities of water when needed was lack of water availability from the source.

Concerning water storage, mothers/caregivers of 79.3% of cases and 67.7% of controls store drinking water in jerry cans. Almost all storage containers were covered during the visit. About 30.8% of cases and 43.4% of controls' mothers cleaned their water storage before filling water. Regarding water drawing from a storage container, 20.2% family members of cases and 32.8% of controls were drawn by dipping can with fingers. Mothers/caregivers of 64.6% cases and 55.1% of controls were washing their hands before water collection. Respondents were also asked whether any household members did anything to make drinking water safer. Accordingly, 83.8% of cases and 76.8% controls' family members informed us that they didn't treat their drinking water to make it safe (Table 2).

Table 2 shows the bivariate conditional logistic regression analysis of childhood diarrhea and water supply related factors. In the analysis, the main sources of water for drinking, cooking and washing, reasons for inability to access sufficient quantities of water when needed, the type of water storage, cleaning of storage before filling, hand washing before water collection and water treatment were significantly associated with childhood diarrhea. But, the reaming variables were not significantly associated with childhood diarrhea. The tables indicate that collecting water mainly from protected well/spring for drinking (crude OR = 1.82; 95% CI: 1.01–3.32), cooking and washing (crude OR = 4.16; 95% CI: 1.47–11.81) purpose were increasing the odds of being having diarrhea compared to those collecting from piped water. Childhood diarrhea was 18% times significantly lower among families storing water in clay pot compared to those families storing in jerry cans (crude OR = 0.18; 95% CI: 0.05–0.61).

A total of 100 (50 from cases and 50 from controls) water samples were collected for bacteriological analysis. The mean pH of the sampled water was 7.03 (±1.01) for cases and 6.67 (±4.94) for controls. Turbidity was also assessed and the mean turbidity of sampled water in cases and controls were 5.36 (±0.68) NTU and 6.59 (±0.70) NTU, respectively. Bacteriological quality of the sampled water was also analyzed in the laboratory and colonies formed were counted. The mean of Total Coliforms was 1859.40 (±1214.88) Colony Forming Units (CFU) per 100 ml of water for cases and 1709.40 (±1074.90) CFU per 100 ml of water for the samples collected from controls' households. The mean of Escherichia coli was 68.90 (±99.99) CFU per 100 ml of water for cases and 81.60 (±107.33) CFU per 100 ml of water for controls. In this study, about 30% of sampled water from cases and 26% sampled water from controls families were free from Escherichia coli, whereas all sampled water analyzed for Total Coliforms showed positive results (Fig 1).

## Sanitation and hygiene factors

About 48% of cases and 55.6% of controls family members defecated in pit latrines with slab whereas the remaining families used unimproved sanitation facilities. Of the latrine facility used by respondents, 58.6% of cases and 61.6% of controls didn't have hand washing facilities. The majority of sanitation facilities of the cases (82.7%) and controls (90.4%) was located in their own yard/plot. Most of these latrine facilities were used by adults all the times of the day and night. Concerning under-five child defecation site, 44.9% of cases and 38.4% of controls practiced open defecation. Half of the mothers/caregivers in this study community disposed of domestic waste in the open. Regarding hand washing, 68.7% of cases and 67.2% of controls' mothers/caregivers reported that they washed their hands at critical times (after defecation,

**Table 2. Water supply related characteristic associated with childhood diarrhea in Kersa and Omo Nada districts of Jimma Zone, Ethiopia.**

| Variables | | Case | Control | Crude OR (95% CI) |
|---|---|---|---|---|
| | | No (%) | No (%) | |
| Main source of drinking-water for members of the household | Piped water | 55(27.8) | 42(21.2) | 1 |
| | Protected well/spring | 139(70.2) | 155(78.3) | 1.82(1.01–3.32)* |
| | Unprotected sources | 4(2.0) | 1(0.5) | 0.41(0.04–3.77) |
| Water available from this source always | yes | 18(9.1) | 16(8.1) | 1 |
| | no | 180(90.9) | 182(91.9) | 1.13(0.57–2.27) |
| Main source of water used by households for cooking and hand washing | Piped water | 34(17.2) | 22(11.1) | 1 |
| | Protected well/spring | 150(75.8) | 167(84.3) | 4.16(1.47–11.81)* |
| | Unprotected sources | 14(7.1) | 9(4.5) | 1.93(0.59–6.36) |
| Distance between drinking water sources and living home approximately (Km) | less than/equal to1 | 192(97.0) | 190(96.0) | 1 |
| | greater than 1 | 6(3.0) | 8(4.0) | 1.5(0.42–5.32) |
| Is the amount of water you get sufficient water? | yes, sufficient | 52(26.3) | 49(24.7) | 1 |
| | Not sufficient | 146(73.7) | 149(75.3) | 1.07(0.70–1.64) |
| main reason for unable to access sufficient quantities of water when needed | water not available from source | 87(59.6) | 103(69.1) | 1 |
| | water too expensive | 18(12.3) | 36(24.2) | 1.18(0.49–2.83) |
| | source not accessible | 41(28.1) | 10(6.7) | 0.17(0.06–0.43)* |
| Drinking water storage containers are observed | clay pots | 31(15.7) | 3(1.5) | 0.18(0.05–0.61)* |
| | Jerry can | 157(79.3) | 134(67.7) | 1 |
| | Buckets | 10(5.1) | 61(30.8) | 6.85(2.39–19.61)* |
| Do the storage container has cover? | yes | 187(94.4) | 194(98.0) | 1 |
| | No | 11(5.6) | 4(2.0) | 1.64(0.77–3.46) |
| Clean storage container before fill | Yes | 137(69.2) | 112(56.6) | 1 |
| | No | 61(30.8) | 86(43.4) | 2.25(1.33–3.81)* |
| way of drawing drinking water from storage container | Pouring | 158(79.8) | 133(67.2) | 1 |
| | Dipping | 40(20.2) | 65(32.8) | 1.58(0.89–2.81) |
| Hand washing before water collection | Yes | 128(64.6) | 109(55.1) | 1 |
| | No | 70(35.4) | 89(44.9) | 2.27(1.23–4.16)* |
| Water treatment | Treat | 32(16.2) | 46(23.2) | 1.82(1.01–3.29) * |
| | Not treat | 166(83.8) | 152(76.8) | 1 |
| Risk of Escherichia coli in sampled water | Very low risk | 15(30) | 13(26) | 1 |
| | Low risk | 3(6) | 3(6) | 1.23(0.14–10.53) |
| | Medium risk | 21(42) | 22(44) | 1.20(0.49–2.97) |
| | High risk | 11(22) | 12(24) | 1.30(0.40–4.23) |

*Statistically significant at P < 0.05

before handling food/water, before feeding a child, after cleaning child stool). Of the interviewed mothers/caregivers, 52.5% of cases and 47% of controls' mothers/caregivers were washing their hands with soap (Table 3).

Bivariate analysis of sanitation and hygiene variables and childhood diarrhea are shown in Table 3. Using this analysis, the sanitation and hygiene risk factors that were significantly associated with childhood were latrine location, latrine use all the times of the day and night, under-five defecation site and domestic waste disposal site (Table 3).

The results of multivariable conditional logistic regression analysis of childhood diarrhea and selected water supply, sanitation and hygiene factors are summarized in Table 4. In the analysis, number of under-fives in the households, wealth status, the main sources of drinking water, hand washing before water drawing from storage container, water treatment, latrine use

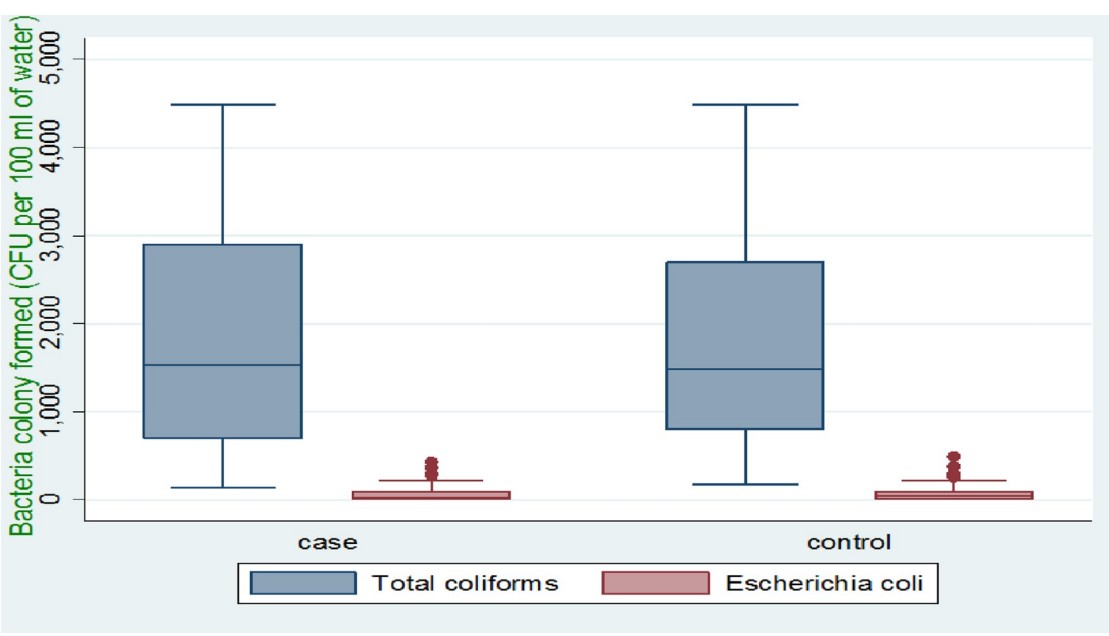

**Fig 1. Colony count of fecal contamination indicator bacteria per 100 ml sampled water of study area.**

**Table 3. Sanitation and hygiene risk factors associated with childhood diarrhea in Kersa and Omo Nada districts of Jimma Zone, Ethiopia.**

| Variables | | Case | Control | Crude OR (95% CI) |
|---|---|---|---|---|
| | | No (%) | No (%) | |
| Place of defecation | Pit latrine with slab | 95(48.0) | 110(55.6) | 1 |
| | Pit latrine without slab/open pit | 96(48.5) | 78(39.4) | 0.68(0.44–1.04) |
| | Bush or field (No facilities) | 7(3.5) | 10(5.1) | 1.17(0.43–3.15) |
| Latrine facility location | In own yard / plot | 158(82.7) | 170(90.4) | 1.94(1.06–3.54)* |
| | Elsewhere | 33(17.3) | 18(9.6) | 1 |
| Presence of hand washing facilities near latrine | Present | 82(41.4) | 76(38.4) | 1 |
| | Absent | 116(58.6) | 122(61.6) | 1.25(0.73–2.14) |
| Latrine use all the times of the day and night (adults) | Yes | 156(81.7) | 176(93.6) | 1 |
| | No | 35(18.3) | 12(6.4) | 0.23(0.10–0.53)* |
| Children under 5 defecation place | Latrine | 43(21.7) | 63(31.8) | 1.99(1.05–3.77)* |
| | Open defecation | 89(44.9) | 76(38.4) | 0.95(0.54–1.66) |
| | Other(poo-poo) | 66(33.3) | 59(29.8) | 1 |
| Child feces disposal | Sanitary | 149(75.3) | 147(74.2) | 1 |
| | Unsanitary | 49(24.7) | 51(25.8) | 1.06(0.67–1.68) |
| Domestic waste disposal site | Household pit | 62(31.3) | 83(41.9) | 4.48(2.00–10.05)* |
| | Dispose in open place | 97(49.0) | 98(49.5) | 2.88(1.40–5.91)* |
| | Burned | 39(19.7) | 17(8.6) | 1 |
| Hand washing at critical times | Yes | 136(68.7) | 133(67.2) | 1 |
| | No | 62(31.3) | 65(32.8) | 1.07(0.71–1.61) |
| Use of soap for wash hands | Use soap | 104(52.5) | 93(47.0) | 1 |
| | Not use soap | 94(47.5) | 105(53.0) | 1.26(0.84–1.87) |

*Statistically significant at P < 0.05

**Table 4. Multivariable conditional logistic regression analysis of childhood diarrhea and associated factors in Kersa and Omo Nada districts of Jimma Zone, Ethiopia.**

| Variables | | Cases No (%) | Controls No (%) | Crude OR (95% CI) | Adjusted OR (95% CI) |
|---|---|---|---|---|---|
| Number of under-five in the household | 1 | 116 (58.6) | 152 (76.8) | 2.5(1.56–4.01) | 2.76(1.33–5.71)* |
| | 2 and above | 82(41.4) | 46(23.2) | 1 | 1 |
| Occupation of mother/caregiver of the child | House wife | 172 (86.9) | 160 (80.9) | 1 | 1 |
| | Merchant | 22(11.1) | 29(14.6) | 1.44(0.79–2.67) | 1.15(0.41–3.25) |
| | Government employee | 4(2.0) | 9(4.5) | 2.39(0.73–7.81) | 1.09(0.16–7.39) |
| Age at child start complementary feeding (Months) | Three | 3(1.5) | 3(1.5) | 1.53(0.29–8.23) | 0.95(0.07–12.53) |
| | Four | 10(5.1) | 21(10.6) | 2.33(1.03–5.24) | 2.40(0.82–7.01) |
| | Six | 169 (85.4) | 165 (83.3) | 1 | 1 |
| | I didn't remember | 16(6.6) | 9(3.5) | 0.53(0.23–1.26) | 0.97(0.26–3.63) |
| Wealth status | Poor | 90(45.5) | 42(21.2) | 1 | 1 |
| | Middle | 57(28.8) | 76(38.4) | 3.28(1.83–5.87)* | 3.69(1.36–10.01)* |
| | Rich | 51(25.8) | 80(40.4) | 3.59(2.01–6.08)* | 5.39(1.99–14.55)* |
| Main source of drinking-water for members of the household | Piped water | 55(27.8) | 42(21.2) | 1 | 1 |
| | Protected well/spring | 139 (70.2) | 155 (78.3) | 1.82(1.01–3.32)* | 4.01(1.40–11.44)* |
| | Unprotected sources | 4(2.0) | 1(0.5) | 0.41(0.04–3.77) | 0.47(0.05–3.88) |
| Main source of water used by households for cooking and hand washing | Piped water | 34(17.2) | 22(11.1) | 1 | 1 |
| | Protected well/spring | 150 (75.8) | 167 (84.3) | 4.16(1.47–11.81)* | 3.69(0.73–14.70) |
| | Unprotected sources | 14(7.1) | 9(4.5) | 1.93(0.59–6.36) | 4.13(0.60–18.43) |
| Clean storage container before full | Yes | 137 (69.2) | 112 (56.6) | 0.44(0.26–0.75)* | 0.78(0.34–1.78) |
| | No | 61(30.8) | 86(43.4) | 1 | 1 |
| Do the storage container has cover? | yes | 187 (94.4) | 194 (98.0) | 1 | 1 |
| | No | 11(5.6) | 4(2.0) | 1.64(0.77–3.46) | 1.33(0.42–4.18) |
| way of drawing drinking water from storage container | Pouring | 158 (79.8) | 133 (67.2) | 0.63(0.36–1.13) | 0.44(0.16–1.24) |
| | Dipping | 40(20.2) | 65(32.8) | 1 | 1 |
| Hands washing before water collection from a storage | yes | 128 (64.6) | 109 (55.1) | 1 | |
| | no | 70(35.4) | 89(44.9) | 2.27(1.23–4.16)* | 4.28(1.46–12.56)* |
| Household members did anything to this water to make it safe | yes | 32(16.2) | 46(23.2) | 1.82(1.01–3.29)* | 1.22(0.48–3.09)* |
| | no | 166 (83.8) | 152 (76.8) | 1 | 1 |
| Place of defecation | Pit latrine with slab | 95(48.0) | 110 (55.6) | 1 | 1 |
| | Pit latrine without slab/open pit | 96(48.5) | 78(39.4) | 0.68(0.44–1.04) | 1.04(0.44–2.47) |
| | Bush or field (No facilities) | 7(3.5) | 10(5.1) | 1.17(0.43–3.15) | - |
| Latrine facility location | In own yard / plot | 158 (82.7) | 170 (90.4) | 1.94(1.06–3.54)* | 2.54(0.95–6.74) |
| | Elsewhere | 33(17.3) | 18(9.6) | 1 | 1 |
| Presence of hand washing facilities near latrine | Present | 82(41.4) | 76(38.4) | 1 | 1 |
| | Absent | 116 (58.6) | 122 (61.6) | 1.25(0.73–2.14) | 1.72(0.66–4.49) |

(*Continued*)

**Table 4.** (Continued)

| Variables | | Cases | Controls | Crude OR (95% CI) | Adjusted OR (95% CI) |
|---|---|---|---|---|---|
| | | No (%) | No (%) | | |
| Latrine use all the times of the day and night | Yes | 156 (81.7) | 176 (93.6) | 1 | 1 |
| | No | 35(18.3) | 12(6.4) | 0.23(0.10–0.53)* | 0.22(0.06–0.78)* |
| Children under 5 defecation place | Latrine | 43(21.7) | 63(31.8) | 1.99(1.05–3.77)* | 0.99(0.28–3.41) |
| | Open defecation | 89(44.9) | 76(38.4) | 0.95(0.54–1.66) | 0.61(0.21–1.80) |
| | Other(poo-poo) | 66(33.3) | 59(29.8) | 1 | 1 |
| Domestic waste disposal | Household pit use | 62(31.3) | 83(41.9) | 4.48(2.00–10.05)* | 4.91(1.39–13.29)* |
| | Disposal in open place | 97(49.0) | 98(49.5) | 2.88(1.40–5.91)* | 5.34(1.70–11.74)* |
| | Burned | 39(19.7) | 17(8.6) | 1 | 1 |
| Use of soap for wash hands | Use soap | 104 (52.5) | 93(47.0) | 1 | 1 |
| | Not use soap | 94(47.5) | 105 (53.0) | 1.26(0.84–1.87) | 2.89(1.35–6.15)* |

*Statistically significant at P < 0.05

all the times of the day and night, domestic waste disposal site and use of soap for hand washing were significantly associated with childhood diarrhea after adjustment.

Children whose mothers/caregivers have one child in their home were 2.76 times more likely to be exposed to diarrhea than children whose mothers had two and above (AOR = 2.76; 95% CI: 1.33–5.71). Children, who started complementary feeding at the age of four months were 2.4 times more likely to develop diarrhea compared to those children started at the age of six months (AOR = 2.40; 95% CI: 0.82–7.01). This association was significant in the bivariate analysis, but after adjustment, their significant association disappeared. In this study, wealth status was also one significant factor. The results of this study also showed that there was increased risk of childhood diarrhea among children whose mothers/caregivers categorized as rich compared to those grouped as poor relatively (AOR = 5.39; 95% CI: 1.99–14.55).

This study showed that exposure to diarrhea among under-five increased by 4 times when their family collected water from protected well/spring than those children whose families collected from piped water facilities (AOR = 4.01; 95% CI: 1.40–11.44). The odds of having diarrhea were lesser among children whose mothers/caregivers cleaned the storage container before filling water than those who didn't clean before full (AOR = 0.78; 95% CI: 0.34–1.78). In addition, the odds of being exposed to diarrhea among children of households who were drawing water from the storage container by pouring was 44% less likely than those drawing by dipping their finger with a can (AOR = 0.44; 95% CI: 0.16–1.24). Children whose mothers/caregivers do not wash hands before water collection were 4.28 times more likely to develop diarrhea than those children who live with mothers/caregivers who wash hands (AOR = 4.28; 95% CI: 1.46–12.56). In this study, water treatment practices was another factor that was significantly associated with childhood diarrhea. We found children living with mothers/caregivers who treated drinking water at home was more likely to suffer from childhood diarrhea compared to children living with mothers/caregivers who didn't practice (AOR = 1.22; 95% CI: 0.48–3.09).

Concerning the sanitation facilities, children whose families defecated in a pit latrine without slab/open pit were 1.04 times more likely to be exposed to diarrhea than children whose families defecated in a pit latrine with slab (AOR = 1.04; 95% CI: 0.44–2.47). Presence of latrine facility in own yard also increased cases of childhood diarrhea compared to those whose latrine is located somewhere out of their compound (AOR = 2.54; 95% CI: 0.95–6.74). The odds of being exposed

to childhood diarrhea were 1.72 times higher in the absence of hand washing facility near latrine compared to the situation in its presence (AOR = 1.72; 95%CI: 0.66–4.49). Children whose families used the pit as a method of waste disposal were 4.91 times more likely to be exposed to diarrhea compared to those whose families burned waste as a disposal method (AOR = 4.91; 95% CI: 1.39–13.29). The likelihood of childhood diarrhea among children from mothers/caregivers who didn't use soap for hand washing was 2.89 times higher than children from mothers/caregivers who used soap during hand washing (AOR = 2.89; 95% CI: 1.35–6.15) (Table 4).

## Discussion

The aim of this study was to examine the association of microbial quality of drinking water and sanitation and hygiene practices on childhood diarrhea in the Kersa and Omo Nada districts of Jimma Zone, Ethiopia. In the study, the number of under-five in the household, wealth status, the main sources of drinking water, hand washing before water drawing from storage container, water treatment, latrine use all the times of the day and night, domestic waste disposal site and use of soap for hand washing were significantly associated with childhood diarrhea.

Unlike previous studies that reported the highest presence of diarrhea in households with more than one under-five children, this study showed that the likelihood of childhood diarrhea were higher in households who have one child. Several numbers of children residing in a household were considered as a predictor of childhood diarrhea in the previous researches [29,30]. But in this study, the presence of more than one child reduced the risk of diarrhea. This might be attributed to poor sanitation and hygienic practices of mothers/caregivers that related to the overall influence of the presence of under-fives in the households.

In addition, the analysis of this study showed that as the wealth status of the respondent increased from poor to wealthy the odds of having diarrhea also increased in a similar fashion. But, previous study findings showed that children living with wealthy family were less exposed to diarrhea because wealth is associated with better access to household facilities related to better hygiene and sanitation and frequently use health services[29,31,32]. Our study finding is inconsistent with the previous findings. A possible reason for this could be that even though the families were wealthy, they may be unable to reduce environmental contaminants which are responsible for childhood diarrhea or the assets and household characteristics we used to generate the wealth categories do not represent wealth accurately [33].

Study findings from Southern Ethiopia and Burundi show that water sources are important environmental predictors of childhood diarrhea [34–36]. Studies identified that changing water source from unimproved to piped water was found to reduce risk of diarrheal disease by 23% [37]. Similarly, in this study, children from households who collected piped water as main sources of drinking water were less likely exposed to diarrhea than those who collected water from protected well/spring. Moreover, using water collected from unprotected (unimproved) sources for cooking/washing or for other domestic purposes was also a risk factor for childhood diarrhea in this study. This study finding is similar to studies done in northwest Ethiopia, Tamale Metropolitan Area of Ghana, South African Villages and Nigeria [32,38–40].

In our study about 70% of sampled water from cases and 74% of sampled water from controls households were contaminated by *Escherichia coli*. Our study finding is inconsistent with a matched case control study done in Addis Ababa slums which reported that stored water contamination by E. coli was found in 83.3% of the case households, and 52.1% of the control households [41]. The difference could be related to the hygienic behavior or socioeconomic difference of study community. The association between the level of Escherichia coli in sampled water and childhood diarrhea has been reported by previous studies [42,43]. In our study, the odds of having diarrhea, increased with the increase of Escherichia coli contamination

level of the sampled water in bivariate analyses. A similar study was reported in a study conducted in Bangladesh [22].

Cleaning water storage container has significantly reduced the risks of diarrhea, but after adjustment, the association disappeared. This might be the confounding association between storage container cleaning and diarrhea. Similar to what have been reported in a study conducted in Rural Peru [44], a way of drawing drinking water from storage container did not significantly associate with the risk of diarrhea in under-fives. However, children living with mothers/caregivers who were drawing drinking water from the storage container by pouring were less likely to develop childhood diarrhea compared to those drawing by dipping their finger with a can. Unlike previous studies [45–47], children living with mothers/caregivers who treat water to make safe were more likely to develop diarrhea than children who were living with mothers/caregivers who didn't treat their drinking water. This might be they didn't properly treat the water, or unsafe handling practices of mothers/caregivers may increase risk of microbial contamination of treated water. This was confirmed by Acharya *et al* [23] who demonstrated that unsafe water handling practices were associated with increasing childhood diarrhea due to microbial contamination.

Improvement in access to improved sanitation facility is one of the important contributors to accelerate the reduction of childhood diarrhea [48]. In the present study, using the unimproved latrine (Pit latrine without slab/open pit) facility increased the risk of childhood diarrhea but the association was not significant. A similar study was reported from Bahir Dar city, Northwest Ethiopia [49]. The study finding, reported from urban India showed that latrine utilization at day and night by adults decreases the probability of open defecation that exposes the residents to diarrheagenic pathogens originating in feces [27]. But, the current study showed that children who live with mothers/caregivers who didn't utilize latrine at day and night were less likely exposed to childhood diarrhea. The inverse association of this latrine utilization at day and night and childhood diarrhea could be attributed to the utilization of unimproved latrine facilities or poor hand washing practices after defecation.

A study showed that inappropriate disposal of domestic waste create breeding sites for insects, which may spread diarrheal pathogens from the open waste to water or food [26,29]. In the present study, children living with mothers/caregivers who were disposing of the domestic waste in households open-pit or open fields were more likely to develop childhood diarrhea compared to those that burn it.

For hand washing after defecation, external factors like availability of hand-washing facilities near latrine facility is important. In this study, around 60% of interviewed mothers/caregivers owned latrine that didn't have a hand washing facility. Moreover, the absence of a hand washing facility near latrine in this study leads children into more risk of diarrhea. The finding of this study is supported by a study done in Yaya Gulele Ethiopia, which indicated that, children from households that had no hand washing facilities were more likely to have diarrhea than the children from households that have hand washing facility near the latrine [46] and a community-based cluster randomized controlled trial study done in Jigjiga District, Eastern Ethiopia [36].

Proper hand washing with soap reduces the concentration of micro-organism close to zero and can interrupt the transmission of childhood diarrhea microbes in the domestic environment [50,51]. This study also showed that children whose mothers/caregivers regularly washed hands with soap were less likely to develop diarrhea compared to those whose mothers/caregivers didn't use soap during hand washing. The finding of this study is consistent with studies done in Ethiopia and Nigeria that warranted hand washing with soap being predictive of lower a likelihood of childhood diarrhea [5,52,53]

Even though the findings of this study will help policymakers to develop comprehensive childhood diarrhea prevention programs in Ethiopia, it has some potential limitations. Firstly,

recall bias and social desirability bias may have occurred due to the individual decision of the mother/caregiver regarding diarrhea and poor reporting of behavioral factors like defecation place, hand washing with soap and child feces disposal practices. Secondly, due to financial problem the amount of water sampled was less and the results were not included in the multivariable conditional logistic regression model.

## Conclusions

Factors that were found to be significantly associated with childhood diarrhea were the number of under-five in the household, wealth status, the main sources of drinking water, hand washing before water drawing from a storage container, water treatment, latrine use all the times of the day and night, domestic waste disposal site and use of soap for hand washing. Therefore, increasing awareness of community and WASH promotion for behavior change was recommended on the water source selection, safe handling and storage water, hand washing practices before water drawing from storage, latrine use and sanitary disposal of domestic waste to prevent childhood diarrhea.

## Supporting information

**S1 File. Questionnaire English.** Questionnaire used for data collection.
(PDF)

## Acknowledgments

We would like to express our deepest appreciations to Kersa and Omo Nada districts health offices, Jimma Zone Health Office and Jimma University Environmental Health department. Special thanks are also extended to study participants.

## Author Contributions

**Conceptualization:** Negasa Eshete Soboksa, Sirak Robele Gari, Bezatu Mengistie Alemu.

**Formal analysis:** Negasa Eshete Soboksa.

**Methodology:** Negasa Eshete Soboksa, Bezatu Mengistie Alemu.

**Supervision:** Sirak Robele Gari, Abebe Beyene Hailu, Bezatu Mengistie Alemu.

**Writing – original draft:** Negasa Eshete Soboksa.

**Writing – review & editing:** Negasa Eshete Soboksa, Sirak Robele Gari, Abebe Beyene Hailu, Bezatu Mengistie Alemu.

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
