## [Decision Letter · Decision Letter 0]

19 Dec 2019

PONE-D-19-25125

Association between microbial water quality, sanitation and hygiene interventions and childhood diarrhea in Kersa and Omo Nadda districts of Jimma Zone, Ethiopia

PLOS ONE

Dear Mr. Saboska,

Thank you for submitting your manuscript to PLOS ONE. After careful consideration, we feel that it has merit but does not fully meet PLOS ONE’s publication criteria as it currently stands. Therefore, we invite you to submit a revised version of the manuscript that addresses the points raised during the review process.

We would appreciate receiving your revised manuscript by 18 January 2020. To enhance the reproducibility of your results, we recommend that if applicable you deposit your laboratory protocols in protocols.io, where a protocol can be assigned its own identifier (DOI) such that it can be cited independently in the future. For instructions see: http://journals.plos.org/plosone/s/submission-guidelines#loc-laboratory-protocols

We look forward to receiving your revised manuscript.

Kind regards,

Russell Kabir, PhD

Academic Editor

PLOS ONE

Journal Requirements:

2. Please state in the Methods:

- Whether the Institutional Review Board (IRB) approved use of oral consent

- How oral consent was documented

For more information, please see our guidelines for human subjects research: https://journals.plos.org/plosone/s/submission-guidelines#loc-human-subjects-research

3. Thank you for including the following funding informagtion within your acknowledgements section; "This work was supported by Addis Ababa University, Ethiopian Institute of Water Resources. The funder had no role in study design, data collection and analysis, decision to publish, or preparation of the manuscript"

"No: The funder had no role in study design, data collection and analysis, decision to publish, or preparation of the manuscript."

Reviewers' comments:

Reviewer's Responses to Questions

**Comments to the Author**

1. Is the manuscript technically sound, and do the data support the conclusions?

Reviewer #1: Yes

Reviewer #2: Yes

2. Has the statistical analysis been performed appropriately and rigorously? 

Reviewer #1: Yes

Reviewer #2: Yes

3. Have the authors made all data underlying the findings in their manuscript fully available?

Reviewer #1: Yes

Reviewer #2: Yes

4. Is the manuscript presented in an intelligible fashion and written in standard English?

Reviewer #1: Yes

Reviewer #2: Yes

5. Review Comments to the Author

Reviewer #1: Introduction: 

Page 2: "A study done in Ethiopia reported that safe child stool disposal is associated with decreased child diarrhea incidences [6]". Comments: To what extent it was decreased, such as 90% or 45%?

Page 2-3: "Obtaining water from shallow wells, where there is overcrowding, poor hygiene and sanitation is practiced, are subjected to contamination with pathogenic organisms [15]." Comments: Please restructure this sentence with grammatical changes, the main statement is unclear here. Also, it would be beneficial to name the pathogenic organisms.

Methods and materials:

Study Design and Population:

Page 4: "The source population was all underfive children living in the two districts."Comments: It will be two separate words of underfive to "under five". Also, it would be great to know the genders of the children here too. 

Results: Water supply related factors:

Comments: In Table 2, it would be great to have an idea about the materials of Pipes, such as lead or PVC?

Figure 1 has to be reshaped and proportionate. 

Major comment for the overall manuscript: The complex sentences should be avoided in most cases and grammatical correction should be done, such as avoiding repetitive words in a single sentence.

Reviewer #2: The authors studied the association between microbial water quality, sanitation and hygiene interventions and childhood diarrhea in Kersa and Omo Nadda districts of Jimma Zone, Ethiopia in a matched case-control study design. However, using the word ‘intervention’ might be a bit misleading to the readers. I would recommend replacing that word with an alternative word such as ‘practices’ or ‘conditions’ in the whole manuscript and the title. Alongside, I have only few minor comments to suggest.

Minor comments:

1. Keep the same font and size for the whole text.

2. Introduction(page3): ‘’There have been significant studies examining the effect of water supply, sanitation and hygiene practices on reducing childhood diarrhoea’’- Please provide references to strengthen the statement.

3. Results (Water supply related factors) (page 8): In the following sentence - ‘Concerning main sources of drinking water, families of 70.2% of cases and 783% of controls collected water from protected well/spring.’, check the percentage for controls.

4. Results (Water supply related factors) (page 8): ‘‘In this study, the average daily water consumption for cases and controls were 12.41 (±5.72) and 13.82 (±6.30) liters respectively’’- Mention clearly if it is cases’ and controls’ family’s daily water consumptions.

5. Page 14: Variables that are adjusted in the multivariable conditional logistic regression analysis in table 4 should also be mentioned as footnote below the table.

6. Lastly, the paper can also benefit from another round of proofreading. There were multiple issues related to grammar, spelling, and word choices.

6. PLOS authors have the option to publish the peer review history of their article (what does this mean?). If published, this will include your full peer review and any attached files.

Reviewer #1: No

Reviewer #2: No

---

## [Author Response · Author response to Decision Letter 0]

8 Jan 2020

As stated in methods part of the ethical clearance section, this study was conducted after Institutional Review Committee of Addis Ababa University, Ethiopian Institute of Water Recourses was approved the use of oral consent. The oral consent was recorded if the study participants agreed to participate in the study after the purpose of the study was explained by data collector before data collection.

Reviewers #1 and Responses:

1. Page 2: "A study done in Ethiopia reported that safe child stool disposal is associated with decreased child diarrhea incidences [6]". Comments: To what extent it was decreased, such as 90% or 45%?

Based on the reviewer’s comments, we have added the % by diarrhea incidence decreased on page 2 (by 23%) 

2. Page 2-3: "Obtaining water from shallow wells, where there is overcrowding, poor hygiene and sanitation is practiced, are subjected to contamination with pathogenic organisms [15]." Comments: Please restructure this sentence with grammatical changes, the main statement is unclear here. Also, it would be beneficial to name the pathogenic organisms.

Thank you for your comments we have rewritten the sentences as follow-on page 3: Poor hygiene and sanitation practices are responsible for the fecal contamination of water obtained from shallow wells.

3. Page 4: "The source population was all underfive children living in the two districts." Comments: It will be two separate words of underfive to "under five". Also, it would be great to know the genders of the children here too.

Thank you but it was written as ‘under-five’. We have included all under five children without sex restriction.

4. Comments: In Table 2, it would be great to have an idea about the materials of Pipes, such as lead or PVC?

We thanks you for good suggestion. By now we consider only microbial contamination of drinking water at point of use. 

5. Figure 1 has to be reshaped and proportionate. It was updated

6. Major comment for the overall manuscript: The complex sentences should be avoided in most cases and grammatical correction should be done, such as avoiding repetitive words in a single sentence.

Reviewers #2 and Responses:

1. The authors studied the association between microbial water quality, sanitation and hygiene interventions and childhood diarrhea in Kersa and Omo Nadda districts of Jimma Zone, Ethiopia in a matched case-control study design. However, using the word ‘intervention’ might be a bit misleading to the readers. I would recommend replacing that word with an alternative word such as ‘practices’ or ‘conditions’ in the whole manuscript and the title.

Based on the reviewer’s comments, we have replace the word ‘interventions’ with ‘practices’ in the whole manuscript and the title. 

2. Keep the same font and size for the whole text.

3. Introduction (page3): ‘’There have been significant studies examining the effect of water supply, sanitation and hygiene practices on reducing childhood diarrhoea’’- Please provide references to strengthen the statement.

We have added the literatures that shows the association of water supply, sanitation and hygiene practices on reducing childhood diarrhoea as recommended in the manuscript on page 3 (cited as 21-24) 

4. Results (Water supply related factors) (page 8): In the following sentence - ‘Concerning main sources of drinking water, families of 70.2% of cases and 783% of controls collected water from protected well/spring.’, check the percentage for controls.

Thank you for your comments. It was changed to 78.3% on page 8

5. Results (Water supply related factors) (page 8): ‘‘in this study, the average daily water consumption for cases and controls were 12.41 (±5.72) and 13.82 (±6.30) liters respectively’’- Mention clearly if it is cases’ and controls’ family’s daily water consumptions.

Thank you. We have rewrite the sentence as follow’ the average daily water consumption for cases were 12.41 (±5.72) liters and for controls were 13.82 (±6.30) liters’.

6. Page 14: Variables that are adjusted in the multivariable conditional logistic regression analysis in table 4 should also be mentioned as footnote below the table.

As the guideline of PLOS ONE journal footnote is not recommended and information about variables that are adjusted in the multivariable conditional logistic regression analysis was written in methods and materials part on page 6 under Data Processing and Analysis.

7. Lastly, the paper can also benefit from another round of proofreading. There were multiple issues related to grammar, spelling, and word choices. 

It was checked through the manuscript.

In addition to the above based on the editor’s comments, we have arranged manuscript according to PLOS ONE's style requirements and removed the funding information from the manuscript and acknowledgment part.

---

## [Editor Report · Decision Letter 1]

4 Feb 2020

Association between microbial water quality, sanitation and hygiene practices and childhood diarrhea in Kersa and Omo Nada districts of Jimma Zone, Ethiopia

PONE-D-19-25125R1

Dear Mr. Soboska,

We are pleased to inform you that your manuscript has been judged scientifically suitable for publication and will be formally accepted for publication once it complies with all outstanding technical requirements.

With kind regards,

Russell Kabir, PhD

Academic Editor

PLOS ONE
---

## [Editor Report · Acceptance letter]

6 Feb 2020

PONE-D-19-25125R1 

Association between microbial water quality, sanitation and hygiene practices and childhood diarrhea in Kersa and Omo Nada districts of Jimma Zone, Ethiopia 

Dear Dr. Soboksa:

I am pleased to inform you that your manuscript has been deemed suitable for publication in PLOS ONE. Congratulations! Your manuscript is now with our production department. 

With kind regards,

on behalf of

Dr. Russell Kabir 

Academic Editor

PLOS ONE